

# A novel immunogenic cell death signature for the prediction of prognosis and therapies in glioma

Jianhua Zhang[1,*], Jin Du[2,*], Zhihai Jin[3], Jiang Qian[1] and Jinfa Xu[1]

[1] Department of Oncology, People's Hospital of Chizhou, Chizhou, China
[2] Department of Neurosurgery, People's Hospital of Chizhou, Chizhou, China
[3] Department of Orthopedics, Handan First Hospital, Handan, China
* These authors contributed equally to this work.

## ABSTRACT

Glioma is a primary cranial malignancy with high recurrence rate, poor prognosis and high mortality. However, the roles of immunogenic cell death (ICD) in glioma remain unclear. Twenty ICD genes were analyzed to be differentially expressed between glioma tissues and non-tumor tissues in 371 glioma patients from The Cancer Genome Atlas (TCGA). Patients were classified into three subgroups *via* unsupervised clustering. Interestingly, the features of cell-infiltrating from three clusters were matched with three immune phenotypes. An applied scoring system was built depending on the expression of hub ICD-related genes. Notably, the ICD-related score was linked with immune checkpoints and the prognosis of glioma patients. In addition, the applied risk model could be used for the prediction of the effect of chemotherapy and immunotherapy for glioma patients. Furthermore, MYD88 was identified to play key roles in the risk model for glioma patients. MYD88 was specifically expressed in malignant cells and validated to correlate with cell proliferation and invasion. Ligand–receptor pairs are determined as novel communications indicating between immunocytes and malignant cells. Therefore, our research established an ICD-related score to investigate the potential effect to chemotherapy and immunotherapy for glioma patients and indicated that MYD88 was a key role in this risk model.

## INTRODUCTION

Gliomas are primary brain tumors that account for 81% of cranial malignancies (*Ostrom et al., 2014*). They are classified as astrocytic, oligodendroglial, and ventricular meningeal tumors according to their histological appearance (*Weller et al., 2015*). They are classified by pathological type into low-grade gliomas (LGG, Grade I-II) and high-grade gliomas (HGG, Grade I-II), also known as glioblastomas (GBM) (*Louis et al., 2016*). GBM are highly aggressive and patients have an extremely poor prognosis, with a 5-year survival

Corresponding authors
Jin Du, HFTS201603@163.com
Jinfa Xu, jinfaxu@163.com

rate of less than 10% and a median overall survival (OS) of approximately 12–17 months (*Stupp et al., 2009*; *Thumma et al., 2012*). Current treatments include surgical resection, radiotherapy, chemotherapy, and immunotherapy, but tumor recurrence and malignant progression still pose a great threat of death to patients (*Lim et al., 2018*). Although surgical resection is the treatment of choices, the infiltrative nature of glioma precludes radical surgery (*Klein, Duffau & De Witt Hamer, 2012*). Surgical treatment combined with adjuvant radiotherapy and chemotherapy adjuvant studies have reported that patients with glioma who received adjuvant chemotherapy had significantly increased survival rates associated with a 2-month increase in median survival time (*Stewart, 2002*). Thus, exploring molecular markers associated with glioma diagnosis and prognosis and investigating relevant novel therapeutic agents are essential to prolong patients' survival.

Without immune activation, induction of cell death by apoptosis is usually regarded as physiological death. If necrosis usually accompanies the onset of an immunogenic inflammatory response, it is referred to as pathological cell death (*Wang et al., 2018*). Therefore, cell death is determined as immunogenic cell death (ICD) by measuring parameters such as the intrinsic antigenicity of the cells, the history of activation or stress prior to cell death, and the nature of the cell death inducer (*Green et al., 2009*). In 2005, for the first study in mice, adriamycin was found to induce ICD in tumor cells (*Casares et al., 2005*). Subsequently, the ICD inducing agent effects of many chemotherapeutic agents such as photodynamic therapy (PDT), various lysing viruses, bortezomib, spautin-1, and shikonin were reported (*Chang et al., 2012*; *Lin et al., 2018*; *Miyamoto et al., 2012*; *Tanis et al., 2015*; *Yang et al., 2018*), and the clinical therapeutic benefits of ICD were confirmed. Clinical studies have shown less toxicity and prolonged OS in glioma patients with low doses of cyclophosphamide (*Schijns et al., 2015*). SRI31277 combined with bortezomib had higher tumor reduction rates relative to its use alone (*Lu et al., 2016*). ICD induced chemotherapy have better prognosis for cancer patients (*Lau et al., 2020*; *Zhou et al., 2019*), but the prognostic impact of ICD on glioma patients is unknown. Moreover, distinct ICD-related prognostic signatures were built and validated to forecast the immune infiltration status of patients with ovarian cancer (*Zhang et al., 2022*) and immunotherapies for patients with breast cancer (*Zhao et al., 2023*) and melanoma patients (*Ren et al., 2022*). Therefore, it is crucial to further investigate the link between ICD genes and glioma.

In this study we investigated the potential roles of ICD genes in the prognosis, immune status and treatment response of patients with glioma. Firstly, RNA sequencing data of patients was collected using publicly available databases to explore the different expression of ICD genes. A consensus cluster analysis was performed to establish ICD subtypes and an applied risk model of ICD-related genes was built. The ICD-related score was associated with the prognosis and could be used to assess the therapeutic effect of individuals after receiving chemotherapy and immunotherapy. MYD88 was identified and validated to regulate cell proliferation and invasion in glioma cells. Thus, our study provided new insights into the role of ICD in glioma and provide new directions for identifying potential biomarkers.

## MATERIALS AND METHODS

### Data source

We searched The Cancer Genome Atlas (TCGA) database to obtain RNA sequencing data of 371 glioma patients and corresponding clinical information and copy number variation (CNV) frequencies of somatic mutations. Gene expression from GSE7696 was extracted to validate the levels of ICD genes in glioma samples (*Murat et al., 2008*). The batch effect was eliminated by the combat algorithm of the "sva" R package. R package "OmicCircos" was applied to identify the locations of genes in the chromosome. We also analyzed an immunotherapy cohort IMvigor210 (*Mariathasan et al., 2018*) to evaluate the possible effect for individuals. To discover the levels of MYD88 in tissues, Human Protein Atlas (HPA) was used to present the immunohistochemistry slides.

### Consensus cluster plus of ICD genes

Thirty-four ICD genes were identified by searching the Molecular Characteristics Database (MSigDB). To investigate the link between ICD genes expression and glioma, we use consensus cluster Plus R package PAM algorithm and "1—Spearman correlation" as distance index ICD genes express the consensus of the clustering analysis. We classified glioma patients into three molecular subtypes according to the different levels of ICD genes. The grouping results were validated using the R package "PCA" (*Shen, Wang & Wu, 2022*). OS was calculated according to the Kaplan–Meier method.

### Gene Set Enrichment Analysis

To identify the biological effects between ICD subtypes, we used a genetic set enrichment analysis website to download glioma related biological pathways and Kyoto Encyclopedia (KEGG) gene and genome data sets. We used gene set variation analysis (GSVA) R package to perform the "ssGSEA" algorithm (*Hanzelmann, Castelo & Guinney, 2013*). We analyzed the enrichment of the three molecular subtypes in different pathways, and the differences were statistically significant when $p < 0.05$.

### Identification of differentially expressed genes (DEGs) between ICD subtypes

In order to determine DEGs, we classified the patients into three subtypes, according to the levels of ICD genes. Glioma patients were divided following an empirical Bayesian approach using the "limma" R package. Two hundred and nine overlapped genes in three subtypes were identified.

### Construction of prognostic models and risk scores associated with ICD

To investigate the survival prognosis of glioma patients, we screened for genes in three molecular subtypes using univariate and multivariate Cox analysis. A columnar line plot was constructed by the "rms" R package to assess the probability of 1-, 3-, 5-year OS. Then, ICD-related score was constructed based on overlapped ICD-related genes. The calculation was performed using the following formula: ICD-related score = $\Sigma$(Exp *
coefi), and patients were further classified into high and low score groups based on median ICD-related score (*Shen et al., 2022*).

## Prediction of chemotherapy drugs

To assess the effect of chemotherapeutic agents in glioma patients, the "pRRophetic" was conducted to analyze half-maximal inhibitory concentration (IC50) values of therapeutic agents from the Genomic of Drugs Sensitivity in Cancer (GDSC) website. The difference was conducted by Wilcox test.

## Cell culture

Human glioma cells LN-229 and U87 were bought from the American Type Culture Collection (ATCC, Manassas, VA, USA). Cells were cultured in Dulbecco's Modified Eagle Medium (DMEM) with 10% fetal bovine serum (Gibco, Billings, MT, CA, USA) at 37 °C with 5% $CO_2$. When 80% cell confluence was attained, cells were routinely passaged.

## RNA extraction and quantitative reverse transcription polymerase chain reaction

Total RNA was extracted from cells using TRIzol reagent (Invitrogen, Waltham, MA, USA) following the manufacturer's protocol. cDNA was obtained by reverse transcription using SuperScript II reverse transcriptase (TIANGEN, Beijing, China), according to the manufacturer's recommended protocol. The specific primer sequences were as follows: MYD88-F: 5′-GGCTGCTCTCAACATGCGA-3′, and MYD88-R: 5′-TGTCCGC ACGTTCAAGAACA-3′, and GAPDH-F: 5′-GGAGCGAGATCCCTCCAAAAT-3′, and GAPDH-R: 5′-GGCTGTTGTCATACTTCTCATGG-3′. The expression of the targeted genes was normalized to GAPDH.

## Single cell RNA-sequencing (scRNA-seq) analysis

ScRNA-seq dataset GSE173278 was downloaded from GEO database (*LeBlanc et al., 2022*). R packages "Seurat" was conducted to normalize the data *via* "normalizationData" function. Cells with 200–10,000 genes, and mitochondrial gene content of ≤10% were chosen for the following analyses. Uniform manifold approximation and projections (UMAPs) were used to show the topmost primary components. Interaction of immune and malignant cells was demonstrated *via* R package 'celltalker', and ligand–receptor pairs were determined.

## Cell transfection

Small interfering RNAs (siRNAs) against MYD88 were bought from GenePharma (Shanghai, China), and the sequence was GCCTATCGCTGTTCTTGAA.

## Cell proliferation and invasion

The methyl thiazolyl tetrazolium (MTT) assay was applied to show cell proliferation. Tumor cells ($5 \times 10^3$) were seeded into 96-well plates and cultured for 24, 48 and 72 h. CellTiter 96 Aqueous One Solution Cell Proliferation Assay kit (Promega, Madison, WI, USA) was added into each well and the absorbance was measured at 490 nm after 4 h

incubation. For the invasion assay, $3 \times 10^3$ cells were added to the upper chamber. After 48 h incubation, cells were fixed with 4% paraformaldehyde and stained with a 0.1% crystal violet.

## Statistical analysis

Mann–Whitney U tests were applied for the analysis between two groups and Kruskal–Wallis tests were used for three groups. The time-dependent area under curve (AUC) was conducted to assess the predictive power of genes for patients' survival. Statistical analyses were conducted with R software (v4.2.0). $p < 0.05$ was considered as significant. ns, no significance. ***$p < 0.001$, **$p < 0.01$, and *$p < 0.05$.

## RESULTS

### Genetic variation of ICD genes in glioma

To analyze genetic mutations and levels of ICD genes in patients, we selected 34 genes associated with ICD and analyzed their mutations in somatic cells and the incidence of CNV. A total of 69 out of 371 samples had mutations in ICD genes, with an incidence of 18.6%. The highest mutation rate was found in PIK3CA (10%) and the mutation rates of CASP1, FOXP3, IFNG, CD88, LY96, P2RX7, and PDIA3 were almost identical at 1% (Fig. 1A). Notably, patients with glioma had a better survival in this nonmutation group than in the mutation group (Fig. 1B). Figure 1C showed the location of CNV alterations in 34 ICD genes on chromosomes. CNV alterations were found in ICD genes with 27 genes showing copy number reduction, while CNV in PIK3CA, P2RX7, CD4, IL17RA, IL10, FOXP3, and CXCR3 amplification frequencies were prevalent (Fig. 1D). We compared the levels of ICD genes between tumor and normal samples from TCGA database. TNF and HSP90AA1 were downregulated in tumor tissues, while ATG5, BAX, CALR, CASP1, CASP8, CD4, CD8B, EIF2AK3, EATPD1, HMGB1, IFNGR1, IL10, LY96, MYD88, NT5E, PDIA3, PRF1 and TLR4 were upregulated in tumor tissues. The expression of CD8A, CXCAR3, FOXP3, IFNA1, IFNB1, IFNG, IL17A, IL17RA, IL1B, IL1R1, IL6, NLRP3, P2RX7, and PIK3CA was not significantly different between tumor tissues and non-tumor tissues (Fig. 1E). Additionally, the levels of ICD genes were also determined in GSE7696 (Fig. S1A).

### Identification of ICD subtypes of glioma

To discover the possible roles of ICD genes, the interaction of ICD genes (Fig. 2A) and the association between ICD genes and immune cells (Fig. 2B) were analyzed. Furthermore, we explored the correlation between 34 ICD genes and glioma clustering by performing consensus cluster analysis on 371 glioma patients from TCGA database. When the clustering variable (k) was set to 3, intra-group correlations were highest and inter-group correlations were lowest, indicating 371 glioma patients could be well classified into three clusters termed cluster A, B and C, based on the ICD genes (Figs. 2C–2E, S1B). We also used a heatmap to compare gene expression differences between the three subtypes (Fig. 2F).

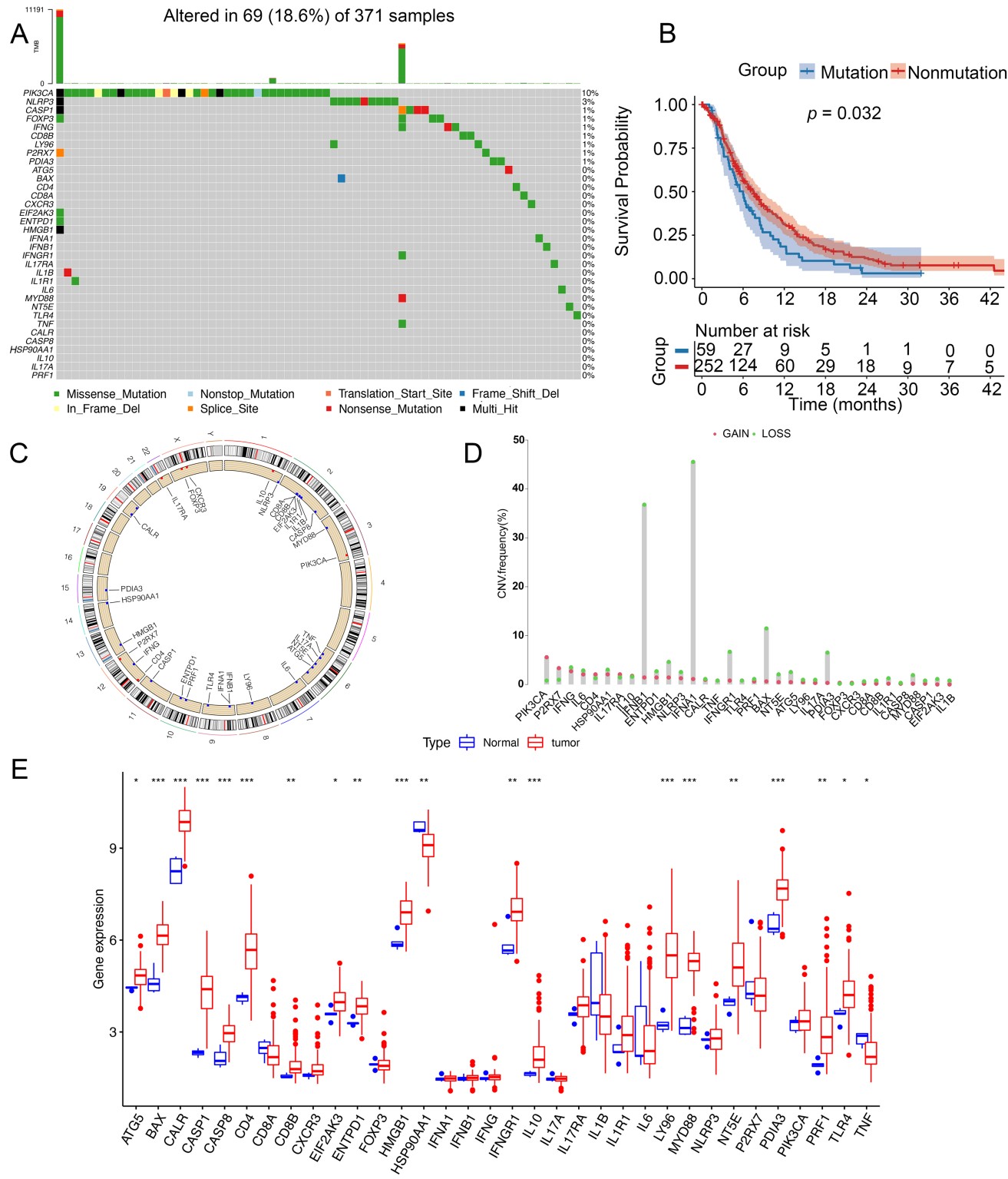

**Figure 1 Genetic changes and expression characteristics of ICD genes in glioma.** (A) Mutations of ICD genes in gliomas. (B) Kaplan–Meier curves showed survival probability of patients with (blue) or without (red) mutations in the ICD-related genes in the TCGA cohort. (C) Genomic location and corresponding expression levels of ICD genes. (D) CNV frequencies of ICD genes. (E) Different expression of ICD genes in normal (red) and tumor tissues (blue) from TCGA. CNV, copy number variation. $^*p < 0.05$, $^{**}p < 0.01$, $^{***}p < 0.001$.

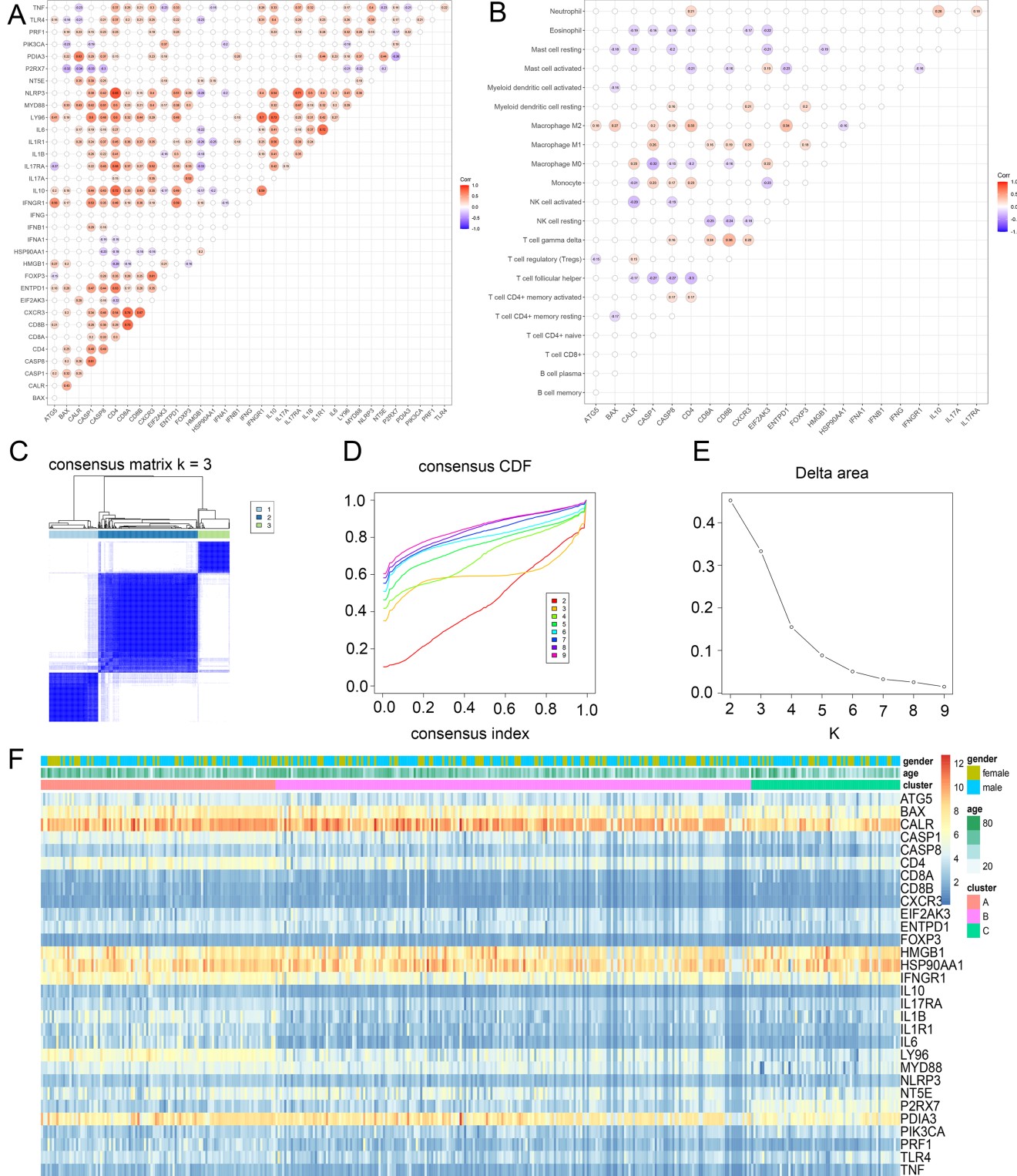

**Figure 2 Identification of ICD subtypes in gliomas.** (A) Consensus clustering of glioma patients with k = 3. (B) Correlation between ICD genes and immune cells. (C) Heatmap of sample clustering under k = 3 in the TCGA database. (D) Consensus clustering cumulative distribution function (CDF) with the number of subtypes k = 2 to 9. (E) Relative change of the area under the CDF curve of k = 2 to 9. (F) Heat map of ICD genes in three clusters. CDF, cumulative distribution function.

## TME cell infiltration features in the subtypes of ICD

Principal component analysis (PCA) results shown an obvious distinction in three clusters (Fig. 3A). Cluster C was examined with a better overall survival than in the cluster A and B (Fig. 3B). We further used GSVA analysis to compare the biological characteristics of three clusters. Different to another clusters, cluster C was enriched in immune activation involving cytokine-cytokine interaction, antigen processing and presentation, B cell receptor signaling pathway and T cell receptor signaling pathway (Figs. 3C and 3D). Moreover, cluster A was correlated with the infiltration of activated immunocytes (Fig. 3E). Combing with a better prognosis, cluster C was identified as an immune-inflamed phenotype with adaptive immune cell infiltration and immune activation. Meanwhile, cluster A was strongly associated with the immune-excluded phenotype that had immune suppression process (Fig. 3C), innate immune cells (Fig. 3E) and TGF-β family member and receptor (Fig. 3F). In addition, cluster B was consistent with the features of immune-desert phenotype including few immune cells and the suppression of the immune response (Figs. 3D and 3F). Thus, these clusters of ICD were analyzed to show different TME features.

## Construction of an ICD-related risk model

To demonstrate the genetic changes based on different ICD clusters, we obtained 209 overlapping DEGs (Fig. 4A). An unsupervised cluster analysis was conducted, and glioma patients were categorized into three subgroups termed genecluster A, B and C. Importantly, a better survival was analyzed in gene cluster C than in other clusters (Fig. 4B). Moreover, a score model for predicting ICD modification in patients was calculated with the levels of ICD-related DEGs. Figure 4C shown the steps of score establishment. It was indicated samples in cluster C (Fig. 4D) and genecluster C (Fig. 4E) had low scores. According to the Wilcoxon test, 15 HLA family genes (Fig. 4F) and 38 immune checkpoints (Fig. 4G) differed in the high- and low-score groups. Therefore, our findings suggested the ICD-related score was linked to tumor immune checkpoints.

## Prediction of the score for chemotherapy and immunotherapy

To explore the chemotherapeutic agents for glioma patients, IC50 of therapeutic agents were determined (Fig. 5A). Patients with low scores were sensitive to Nutlin.3a (Fig. 5B) and vorinostat (Fig. 5C) while those in the high-score group exhibited strong sensitivity to BX.795 (Fig. 5D) and BAY.61.3603 (Fig. 5E), determining the score was considered as a predictive biomarker of chemotherapy for glioma patients. Next, the prediction of the score for immunotherapy was also demonstrated. Patients with low scores had better prognostic conditions for anti-PD-L1 (Fig. 5F). Patients with low scores enjoyed significant treatment effects and enhanced anti-PD-L1 immunoreactivity (Fig. 5G). In IMvigor210, favorable effect of treatment and immune responsiveness to the PD-L1 blockade was indicated in the group with low scores (Figs. 5G and 5H). In addition, patients with low scores and low neoantigen load benefited in the survival (Fig. 5I). The low scores were linked to the inflamed immune phenotype (Fig. 5J). Thus, the score was linked to tumor immune phenotypes and applied in the prediction of anti-PD-L1 immunotherapy.

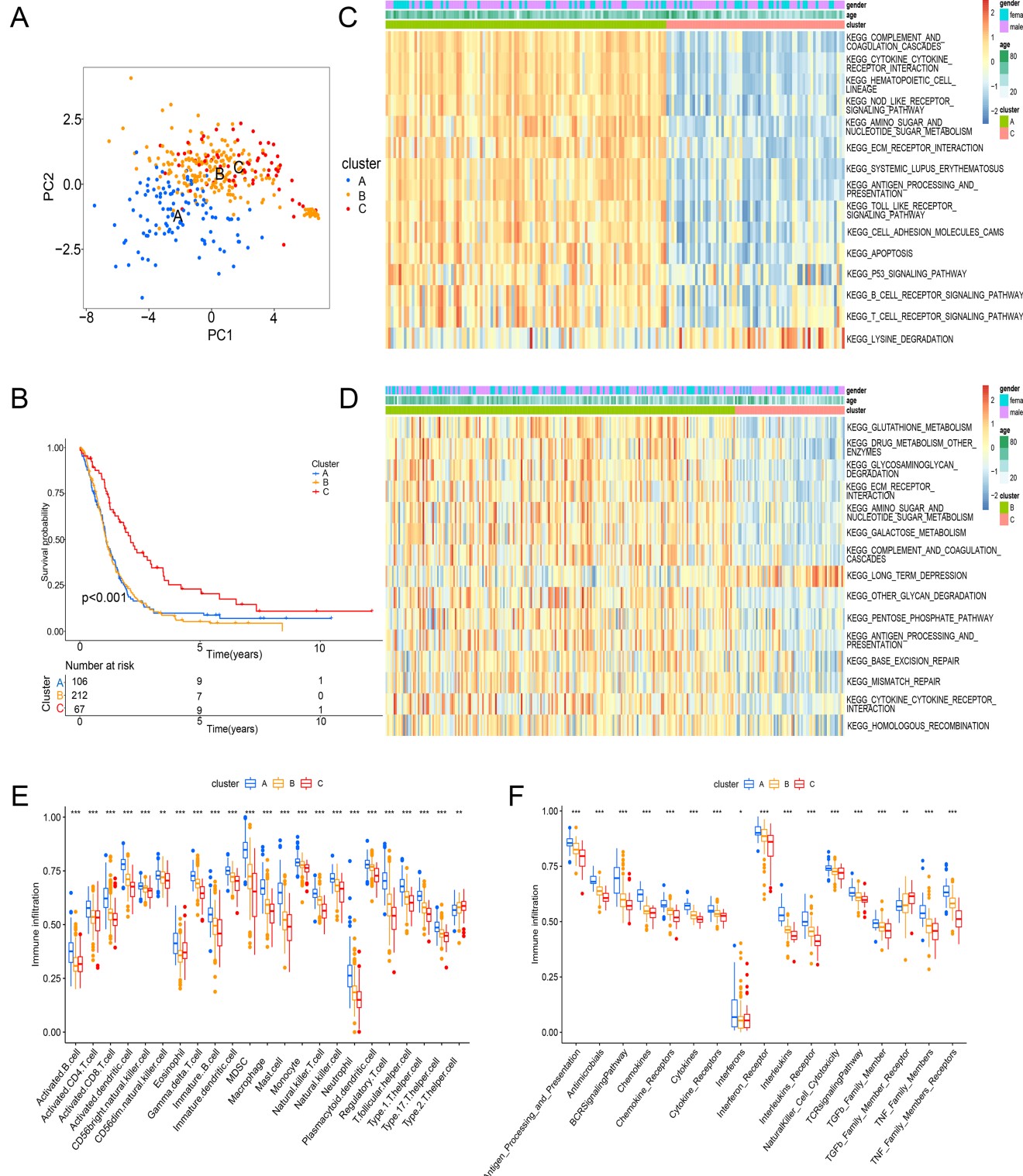

**Figure 3 Enrichment analysis and immune infiltration of different ICD subtypes.** (A) PCA analysis showed significant differences in gene expression of ICD subtypes. (B) Kaplan–Meier curves and significant differences in survival rates among the three subtypes. (C and D) GSVA enrichment analysis showed activation of various pathways under different patterns of ICD modifications. (E) Difference of immune infiltrating cells in three clusters. (F) Characteristics of immune responses in different clusters. PCA, principal component analysis; GSVA, gene set variation analysis. $*p < 0.05$, $**p < 0.01$, $***p < 0.001$.

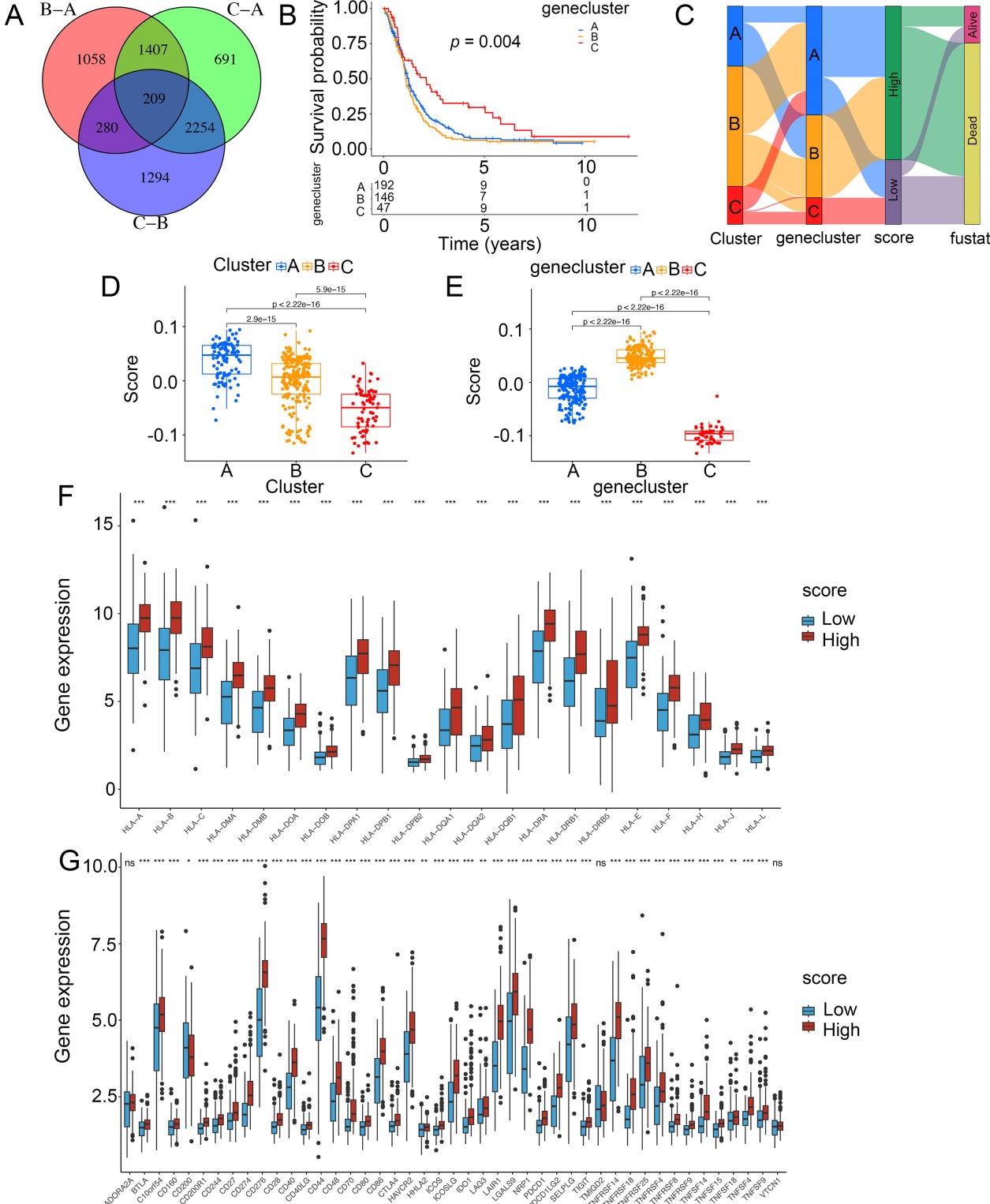

**Figure 4 Construction of ICD-related score in glioma.** (A) A total of 209 ICD-related genes shown in Venn diagram. (B) Kaplan–Meier curves of survival in the glioma cohort with three distinct geneclusters. (C) Alluvial diagram showing the changes in clusters, gene clusters and scores. Scores in distinct (D) clusters and (E) geneclusters. Analyses for (F) the expression of HLA family genes and (G) immune checkpoints in the score groups. *$p < 0.05$, **$p < 0.01$, ***$p < 0.001$, ns, not significant.

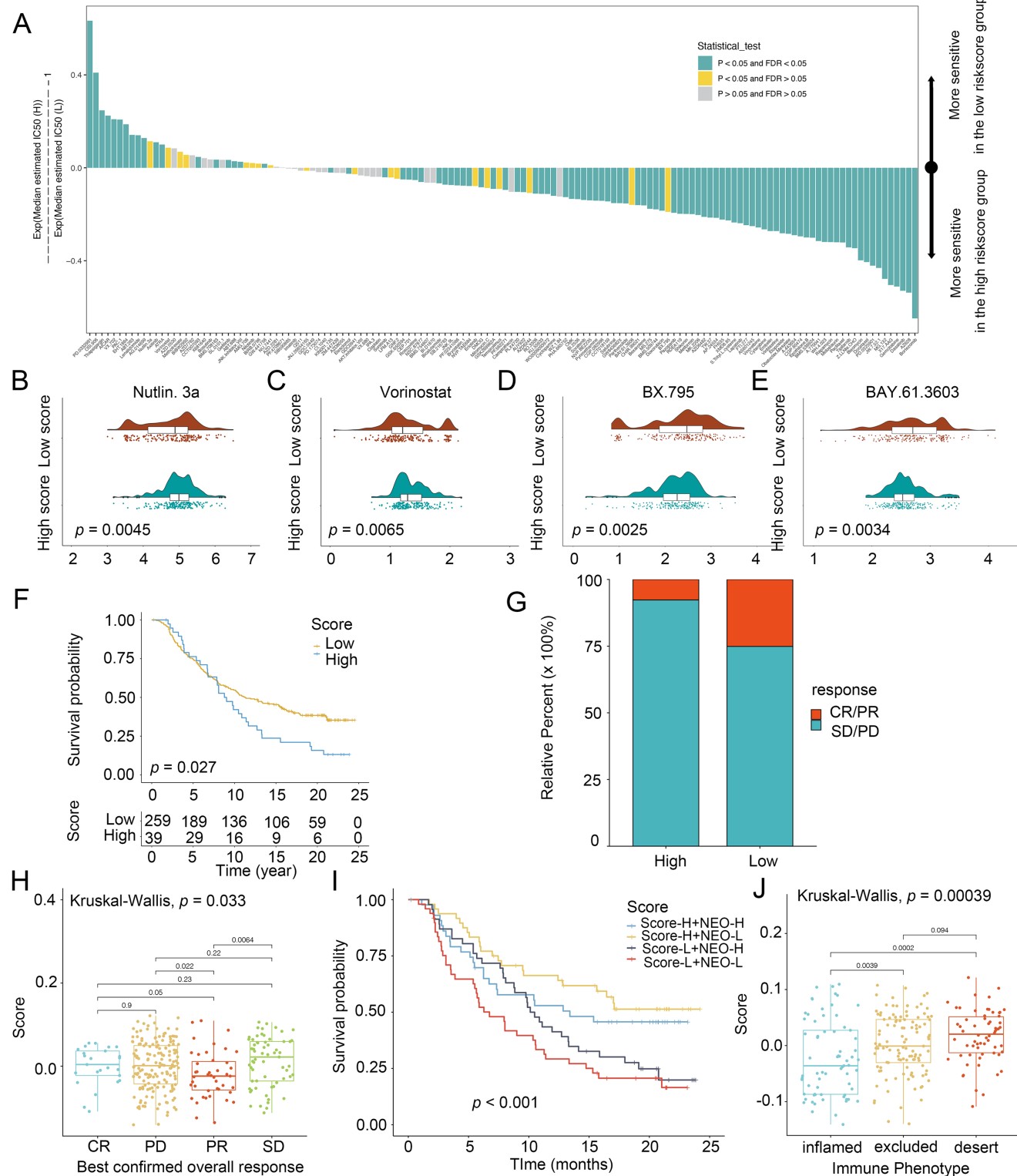

**Figure 5 Score in the prediction of chemotherapies and immunotherapies.** (A) Sensitivity of 138 drugs. The efficacy of (B) Nutlin. 3a, (C) Vorinostat, (D) BX.795 and (E) BAY.61.3603 for high- and low-score groups. (F) Kaplan–Meier curves of survival in the high- and low-score groups in the patients receiving anti-PD-L1 therapy. (G) Proportion of patients with response to PD-L1 blockade immunotherapy in the high and

**Figure 5 (continued)**
low-score groups. (H) Distribution of scores in distinct anti-PD-L1 clinical response groups. (I) Survival analyses for patients receiving anti-PD-L1 immunotherapy stratified by the combination of score and neoantigen burden. (J) Differences in scores among distinct tumor immune phenotypes in IMvigor210 cohort. SD, stable disease. PD, progressive disease. CR, complete response. PR, partial response. NEO, neoantigen burden.

## Clinical values of the score in glioma

To identify key ICD genes in this risk model, 16 DEG-based optimal prognosis related genes were found by univariate Cox analysis. The results showed that P2RX7 was a protective gene with a hazard ratios (HRs) <1, and 15 risk genes (CALR, CASP1, CASP8, CD4, CXCR3, FOXP3, IL10, IL17RA, IL1B, IL1R1, IL6, LY96, MYD88, PDIA3, and PRF1) with HRs >1 (Fig. 6A). In addition, we performed multivariate Cox analysis and identified three key genes FOXP3, MYD88, and IL6 (Fig. 6B). To assess the survival probability in glioma patients, a nomogram was built to calculate the probability of 1-, 3-, 5-year OS (Fig. 6C). Time-dependent AUC curves indicated that MYD88 acted as a predictive biomarker for patients' survival (Fig. S2). Strikingly, we found that MYD88 was identified to associate with OS (Fig. 6D) and progression free interval (PFI) (Fig. 6E) in a range of cancer.

## Specific expression of MYD88 in glioma cells

To discover the function of MYD88 in various types of cancer, we compared the expression of MYD88 in tumor and non-tumor tissues. The levels of MYD88 in tumor samples were much higher than in the non-tumor samples in nine types of cancer including glioma (Fig. 7A). Moreover, to investigate the expression of MYD88 in glioma, we explored the levels of MYD88 in several cell types in malignancies by scRNA-seq. After quality control, a whole-transcriptome database of 29,543 cells from 10 glioma patients was analyzed. Based on cell-specific markers, six cell types (endothelial, fibroblast, immune, malignant, neuron and oligodendrocyte) were determined (Fig. 7B). The proportion of malignant cells was high in the patients (Fig. 7C). Notably, the expression levels of MYD88 were enriched in malignant cells, suggesting a specific expression in malignancies (Fig. 7D). The characteristics of glioma patients with different MYD88 expression from TCGA were shown in Table 1. HPA indicated MYD88 protein levels in glioma tissues were higher than in normal tissues (Fig. 7E). Moreover, KEGG analysis determined that the biological functions of MYD88 were enriched in MYC targets, protein secretion and oxidative phosphorylation (Fig. 7F). All above data indicated that MYD88 was specifically expressed in glioma cells.

## Effect of MYD88 knockdown on cell proliferation and invasive ability

To determine the roles of MYD88 in glioma, scramble (scr) and siRNA for MYD88 were transfected into glioma cells (LN-229 and U87). After the transfection of siRNA, the levels of MYD88 were significantly downregulated in LN-229 and U87 cells (Fig. 8A). The MTT assay indicated the proliferative rate of LN-229 and U87 cells was significantly downregulated upon MYD88 knockdown after 72 h (Fig. 8B). In addition, the transwell

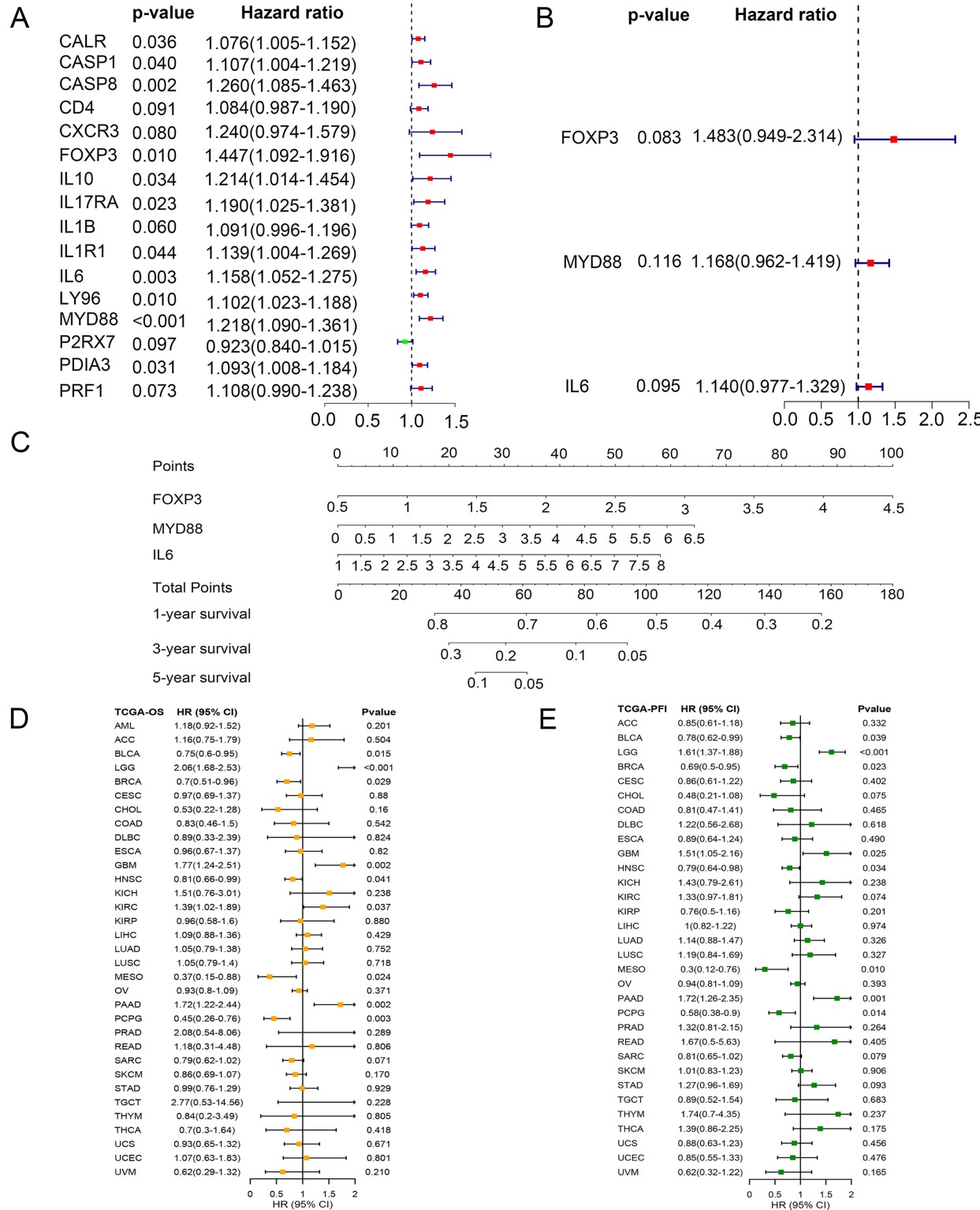

**Figure 6 Prognostic values of MYD88 in pan-cancer.** (A) Univariate Cox analysis and (B) multivariate Cox analysis was used to evaluate the prognostic value of key genes for glioma patients. (C) Nomogram predicting 1-, 3- and 5-year OS in glioma patients. (D) Overall survival of patients with high levels of MYD88 in pan-cancer. (E) Progression free survival of patients with high levels of MYD88 in pan-cancer.

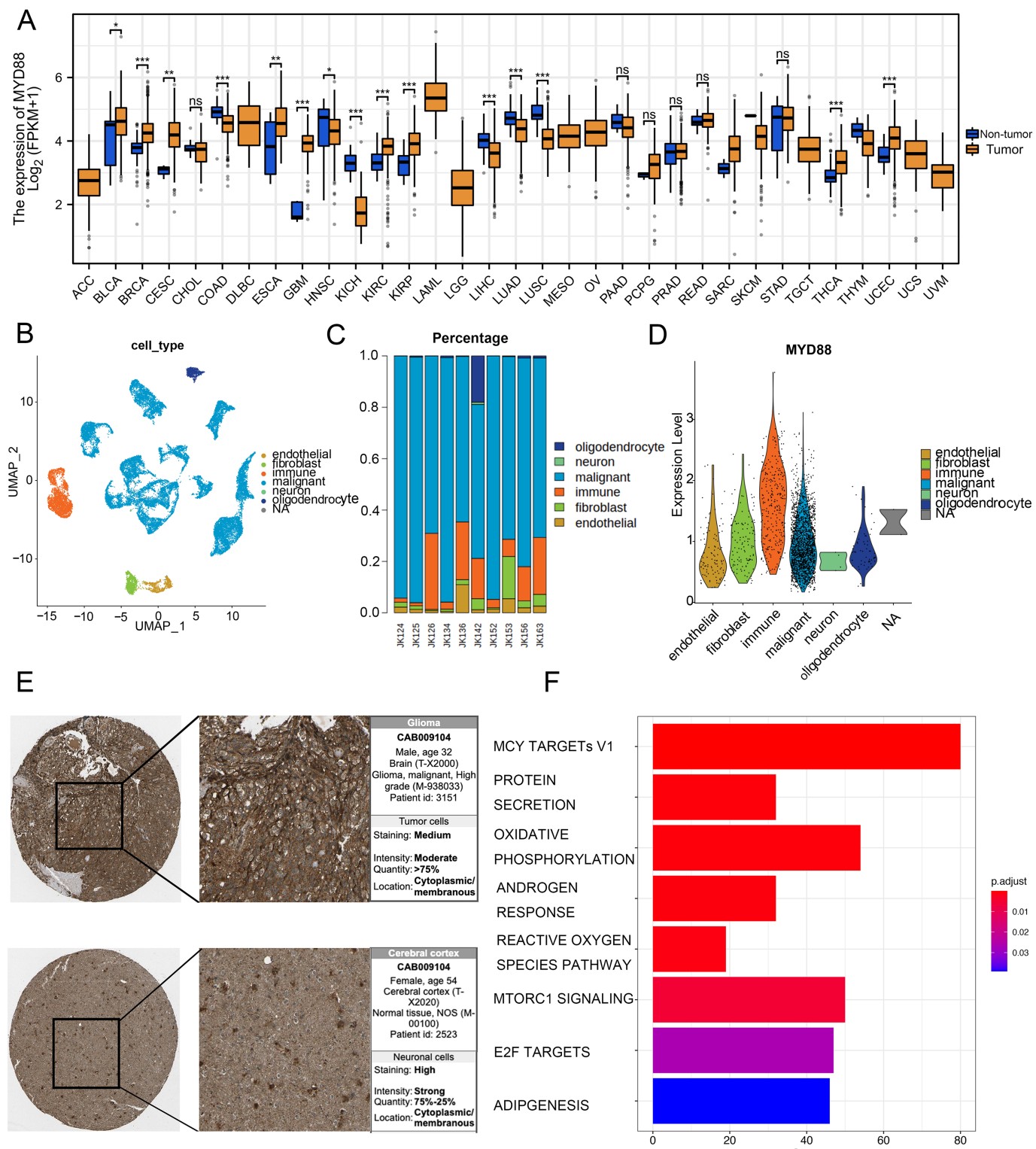

**Figure 7 Specific expression of MYD88 in glioma cells.** (A) Levels of MYD88 in 33 types of cancer. (B) UMAP indicated the cell composition in the microenvironment of glioma according to cell types. (C) Proportion of cell types in patients. (D) Levels of MYD88 in cell types. (E) Protein levels of MYD88 by immunohistochemical staining analysis based on the Human Protein Atlas database (10×). (F) KEGG shown the biological functions of MYD88. UMAP, uniform manifold approximation and projection. $^*p < 0.05$, $^{**}p < 0.01$, $^{***}p < 0.001$, ns, not significant.

Table 1 Characteristics of glioma patients from the TCGA database.

| Characteristics | Low expression of MYD88 | High expression of MYD88 | p value |
|---|---|---|---|
| n | 84 | 84 | |
| Gender, n (%) | | | 0.628 |
| Female | 28 (16.7%) | 31 (18.5%) | |
| Male | 56 (33.3%) | 53 (31.5%) | |
| Age, n (%) | | | 0.877 |
| <=60 | 44 (26.2%) | 43 (25.6%) | |
| >60 | 40 (23.8%) | 41 (24.4%) | |
| IDH status, n (%) | | | 0.068 |
| WT | 71 (44.1%) | 78 (48.4%) | |
| Mut | 9 (5.6%) | 3 (1.9%) | |
| Karnofsky performance score, n (%) | | | 0.432 |
| >80 | 44 (34.4%) | 48 (37.5%) | |
| <80 | 20 (15.6%) | 16 (12.5%) | |
| OS event, n (%) | | | 0.694 |
| Alive | 17 (10.1%) | 15 (8.9%) | |
| Dead | 67 (39.9%) | 69 (41.1%) | |
| DSS event, n (%) | | | 0.516 |
| No | 19 (12.3%) | 15 (9.7%) | |
| Yes | 60 (38.7%) | 61 (39.4%) | |
| PFI event, n (%) | | | 0.238 |
| No | 19 (11.3%) | 13 (7.7%) | |
| Yes | 65 (38.7%) | 71 (42.3%) | |

assay determined that MYD88 knockdown reduced cell invasion in LN-229 and U87 cells (Fig. 8C). The findings suggested the oncogenic roles of MYD88 in glioma.

## Ligand–receptors pairs between immunocytes and glioma cells with high levels of MYD88

We explored the interaction between immunocytes and glioma cells by scRNA-seq analysis. Glioma cells with high MYD88 levels interacted with macrophage, naïve T cell, microglial cell, and dendritic cell actively. In this study, cells with high MYD88 levels could communicate with naïve T cell, macrophage, dendritic cell through ANGPTL pathway (ANGPTL2-TLR4, Fig. 9A), GAS pathway (GAS6-AXL, Fig. 9B), GRN pathway (GRN-SORT1, Fig. 9C), LIGHT pathway (TNFSF14-LTBR, Fig. 9D), LT pathway (LTA-TNFRSF1A, Fig. 9E) and VEGF pathway (PGF-VEGFR1, Fig. 9F).

## DISCUSSION

Glioma is the most common primary cranial malignancy, and the prognosis of glioma patients is not ideal because radical surgery is not possible. Clinical studies have found that ICD induced chemotherapy regimens can prolong the survival of cancer patients and may provide a new option for the treatment of glioma (Wang et al., 2018). We identified the

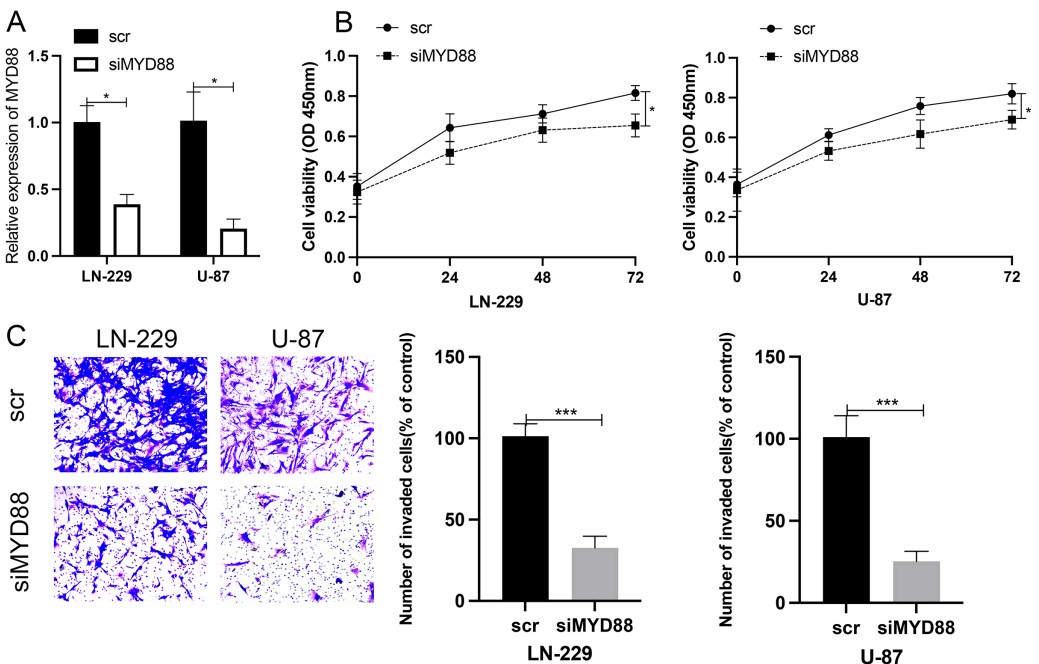

**Figure 8 Knockdown of MYD88 inhibited the proliferation and invasion of glioma cells.** (A) qPCR showed the relative levels of MYD88 after transfection of siRNA. (B) MTT assay of the proliferation of LN-229 and U87 cells after 24, 48 and 72 h. (C) Transwell cell assays demonstrated cell invasion with crystal violet in siMYD88-transfected LN-229 and U87 cells compared with non-specific scramble (scr) siRNA as a control group, 10×. siRNA, small interfering RNA. $^*p < 0.05$, $^{***}p < 0.001$.

distinct expression of ICD genes between tissues and non-tumor tissues. Considering the tumor heterogeneity of glioma, we conducted consensus clustering analysis on ICD genes to obtain three clusters and initially explored the differences in biological characteristics, survival, and immune infiltration among subtypes. A novel scoring system was built to assess the chemotherapeutic drugs and immunotherapy for the individuals. MYD88 was considered as prognostic biomarker for glioma patients. Moreover, MYD88 was associated with cell proliferation and invasion. Thus, we established an ICD-related score to assess the glioma patients for chemotherapy and immunotherapy.

In recent years, dozens of chemotherapeutic agents have been found to be considered as ICD inducers (*Martins et al., 2011*; *Tesniere et al., 2010*), and combination therapy for ICD induction significantly improves OS in patients with gastrointestinal, lung, and cervical cancers (*Hijikata et al., 2018*; *Murahashi et al., 2016*). In our study, we identified 20 differentially expressed ICD genes between glioma tissues and non-tumor tissues. The interaction in ICD genes was found to be associated with immune cells. Furthermore, the glioma patients were classified into three ICD clusters. Strikingly, three clusters were matched with the cell-infiltrating characteristics of three immune phenotypes of tumors, which is consistent with another study (*Shen et al., 2022*). Cluster C was correlated with immune activation and prolonged survival, consistent with the feature of immune-inflamed phenotype. Based on the levels of overlapped genes from genecluster, we established an applied scoring system to assess the glioma patients. The risk score

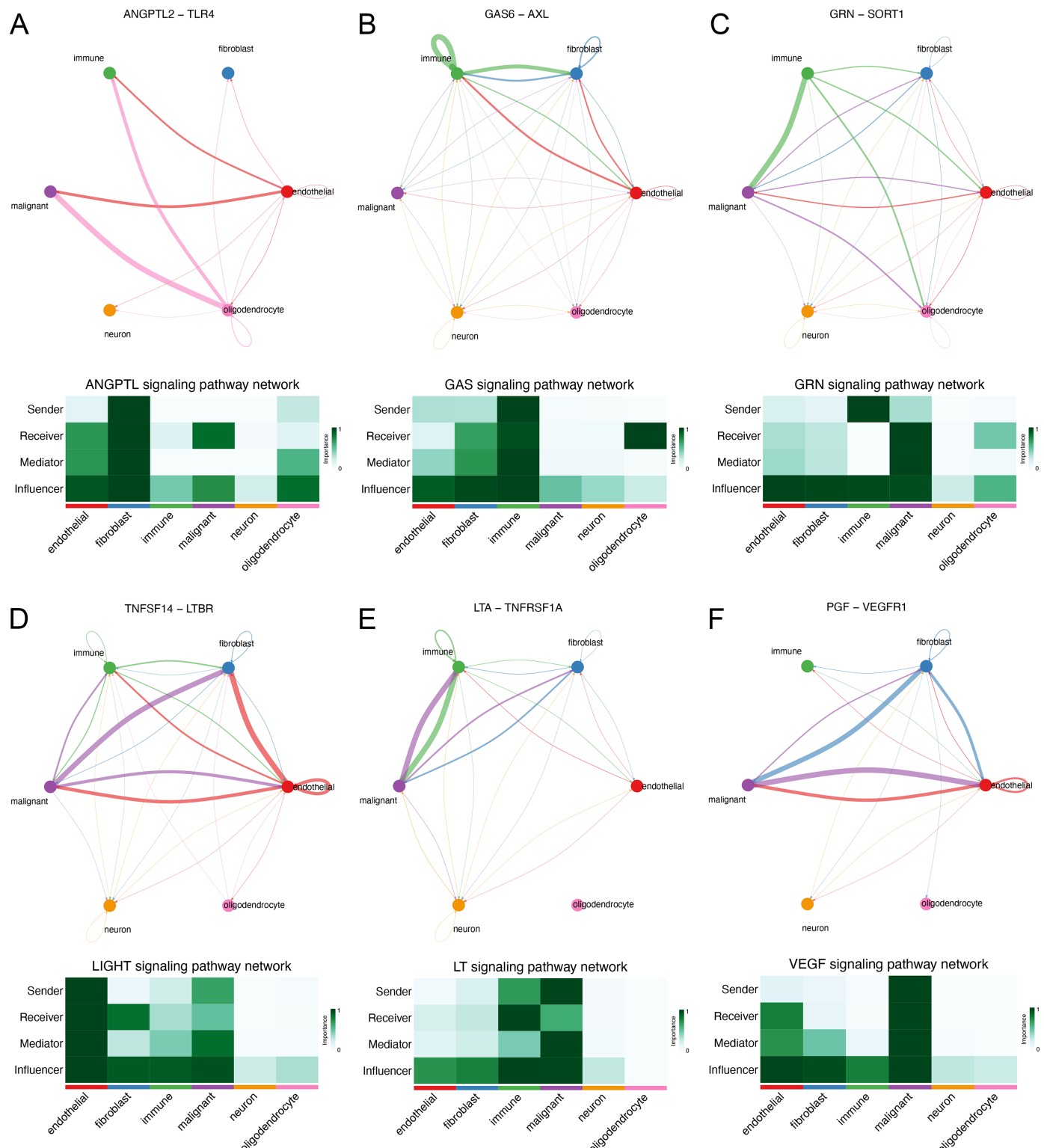

**Figure 9  Novel ligand–receptor pairs difference between high and low MYD88 patients.** (A) High MYD88 cells communicated with endothelial, oligodendrocyte, and immune cells *via* ANGPTL2-TLR4. (B) High MYD88 cells communicated with endothelial, fibroblast and immune cells through GAS6-AXL. (C) High MYD88 cells communicated with oligodendrocyte and immune cells through GRN-SORT1. (D) High MYD88 cells communicated with endothelial and fibroblast *via* TNFSF14-LTBR. (E) High MYD88 cells communicated with fibroblast and immune cells *via* LTA-TNFRSF1A. (F) High MYD88 cells communicated with endothelial and fibroblast *via* PGF-VEGFR1.

system has been found in gastric cancer (*Liu et al., 2021*), breast cancer (*Xu et al., 2022*) and another malignancy (*Pal Choudhury, Chaturvedi & Chatterjee, 2020*). In this study, the score was explored to be strongly associated with immune checkpoints. Moreover, the predictive values of this score were validated in patients receiving immunotherapy. Patients with low scores had prolonged survival than ones with high scores after anti-PD-L1 therapy, validating the assessment of this score for glioma patients receiving immunotherapy. We also explored the evolution of the score for chemotherapy. Vorinostat, an inhibitor of class I and II of histone deacetylases (HDACs), was identified to be sensitive for glioma patients with low scores. Meanwhile, vorinostat was identified to enhance the anticancer effects in combination with other anticancer drugs in small cell lung cancer (*Pan et al., 2016*) and breast cancer (*Wawruszak et al., 2021*). Thus, the applied scoring system could be used for the prediction of chemotherapy and immunotherapy for personalized treatment.

We constructed a univariate and multivariate cox regression analysis based on ICD-DEGs to identify three genes (FOXP3, MYD88 and IL6) associated with prognosis of glioma patients. MYD88, a myeloid differentiation primary response protein, was determined to correlate with OS and PFI, suggesting a potential prognostic value in various types of cancer. Furthermore, it has been found to mediate cell proliferation, migration and invasion in colorectal cancer (*Zhu et al., 2020*) and non-small cell lung cancer (*Xiang et al., 2014*). The levels of MYD88 were higher in tumor samples in nine types of cancer than in non-tumor tissues, especially in glioma. Furthermore, strong expression of MYD88 from clinical samples of gastric cardia carcinoma (*Chen et al., 2020*) supported our finding and suggested an oncogenic role of MYD88 in cancer. Meanwhile, low levels of MYD88 from the cohort of colorectal cancer was consistent with our results (*Li et al., 2014*). By single cell RNA sequencing analysis, we could find that MYD88 was specifically expressed in malignant cells not in other cell clusters, enhancing the oncogenic roles of MYD88 in glioma. Immunohistochemistry (IHC) from HPA supported the finding that the protein levels of MYD88 were higher in glioma samples that in normal tissues. The possible pathways that MYD88 was involved in provided the roles of cell growth and apoptosis. To confirm the reliability and accuracy of the results of the above bioinformatics analysis, the cell proliferation and invasion ability after MYD88 knockdown were determined by siRNA transfection. The data supported the oncogenic role of MYD88 in glioma, consistent with the findings in colorectal cancer and breast cancer (*Wu et al., 2018*; *Zhu et al., 2020*). Notably, we also explored the possible ligand–receptor pairs between malignant and immune cells to discover the potential interactions. The interaction of GAS6-AXL has been found in glioma (*Xiang et al., 2020*) while we identified several novel combinations in our study which need to be validated by the experiments in glioma.

Furthermore, some limitations existed in our study. The predictive significance of the ICD-related score obtained needs to be confirmed by large-scale prospective clinical studies. In addition, the potential value of ICD needs to be explored in more studies (*Franco-Molina et al., 2021*).

## CONCLUSIONS

ICD has important implications for tumor cell death and cancer treatment. In the present study, we found that ICD-related score could be considered as a biomarker for the prognosis and the prediction of chemotherapeutic agents and immunotherapy for glioma patients. Moreover, MYD88 was identified to associate with cell proliferation and invasion. Therefore, our study provided new avenues of research on the role of ICD and explored new biomarkers to guide the diagnosis and therapy for glioma patients.

## ACKNOWLEDGEMENTS

All authors are grateful to all the patients who contributed data to this study.

### Funding

The authors received no funding for this work.

### Competing Interests

The authors declare that they have no competing interests.

### Author Contributions

- Jianhua Zhang performed the experiments, analyzed the data, prepared figures and/or tables, authored or reviewed drafts of the article, and approved the final draft.
- Jin Du conceived and designed the experiments, authored or reviewed drafts of the article, and approved the final draft.
- Zhihai Jin performed the experiments, analyzed the data, prepared figures and/or tables, authored or reviewed drafts of the article, and approved the final draft.
- Jiang Qian performed the experiments, analyzed the data, prepared figures and/or tables, authored or reviewed drafts of the article, and approved the final draft.
- Jinfa Xu conceived and designed the experiments, authored or reviewed drafts of the article, and approved the final draft.

### Data Availability

The datasets analyzed for this study are available at: TCGA-LUAD (http://www.cancer.gov/tcga); GEO: GSE7696, GSE173278, and IMvigor210 (https://clinicaltrials.gov/ct2/show/NCT02108652).

https://www.ncbi.nlm.nih.gov/geo/query/acc.cgi?acc=GSE7696.

https://www.ncbi.nlm.nih.gov/geo/query/acc.cgi?acc=GSE173278.

### Supplemental Information

Supplemental information for this article can be found online at http://dx.doi.org/10.7717/peerj.15615#supplemental-information.

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
