# Peer review of "A novel immunogenic cell death signature for the prediction of prognosis and therapies in glioma"

_PeerJ, doi:10.7717/peerj.15615_

## Round 0.1 · original submission · Major Revisions

It requires a number of Major Revisions.

Reviewer 1 ·

Basic reporting

Language is clear, intelligible and professional.
Literature references, sufficient field background are well provided.
Good article structure, figures, tables. Raw data are provided.

Experimental design

The topic of this manuscript is matched with the aims of this journal.
Research question is well defined and meaningful.
Rigorous investigation and validation were performed.
Methods were described clearly and easy to replicate.

Validity of the findings

It is a novelty to validate clinical significance of the riskscore based on ICD-related genes in glioma.
All data is provided and controlled.
The results and conclusions are well supported by the data in this manuscript.

Additional comments

This manuscript is well written and contributes a new direction scheme for the therapy of glioma. In this manuscript, the authors have investigated the roles of ICD in glioma and established a riskscore of ICD for the prognosis and prediction of chemotherapy and immunotherapy for glioma patients. Interestingly, three clusters of ICD were consistent with three immune phenotypes. In addition, single cell RNA sequencing analysis highlighted the specific expression of MYD88 in malignant cells. The clinical values of this riskscore were well validate in the prediction of chemotherapy and immunotherapy. Overall, the authors have fully elucidated their results.

Meanwhile, there are some comments for improving the manuscript.
1. The expression of ICD-related genes needs to be validated in GEO datasets.
2. Contents in figure 1 to 3 are not mismatched with figure legends. Please make sure everything is correct.
3. The abstract should be improved.
4. It is necessary to highlight the clinical significance of ICD-related signature for the prognosis, diagnosis and therapy in other cancer. Line 61 – 64.
5. The bars in figure 8a are missing and the name of cell line in the right panel in figure 8B seems to be wrong. It needs to be checked.
6. The p value is missing in figure 5I.
7. The authors described the roles of MYD88 in various types of cancer in figure 6D, E and 7A. The results can be discussed in the discussion.

Reviewer 2 ·

Basic reporting

The language in this manuscript is clear and easy to understand. The structure is matched with PeerJ standards. Intro shows the background; the methods are described while results and conclusions are well confirmed by the data. High resolution figures are shown, and a table is well labelled. Raw data is supplied.

Experimental design

This research meets the scope of PeerJ. The topic of this research is meaningful and shows a potential clinical values for glioma patients. Necessary investigations are performed to support the conclusions. Methods are described with sufficient details.

Validity of the findings

All data shown in the manuscript are supplied and they are clear and stable to replicate. Notably, the authors explored an applied ICD-related riskscore to predict the effect of immunotherapy and chemotherapy for glioma patients, indicating a novelty in the field of glioma. Furthermore, the conclusions are well supported by the data and results in this research.

Additional comments

From public gene expression datasets, the authors identified a set of differentially expressed ICD genes for risk stratification and they built a riskscore based on the expression of ICD-related genes. The clinical application of the riskscore was determined and it could be used for the assessment of chemotherapeutic agents and immunotherapy. It is validated by single cell RNA sequencing analysis from public datasets to indicate the novel roles of MYD88 in glioma. The basic experiments supported the important roles of MYD88 in the proliferation and invasion. Thus, this research well identified an ICD-related score to investigate the potential effect for chemotherapy and immunotherapy for glioma patients.
Strengths and limitations
The conclusions of this research were well supported by the results and raw data. Notably, single cell RNA sequencing analysis also shown the specific expression of MYD88 in malignant cells, suggesting a novel role in glioma. The clinical application of the riskscore were assessed, and it provided the possibility of personalized treatment for every glioma patient. Meanwhile, there are some limitations to revise.

1. I highly recommend revising the abstract and make sure it is unambiguous and professional.
2. The work provides a very broad pipeline of tools addressed at the discovery of glioma biomarkers. The main limitation is the lack of detail on some of the computational methods. More parameter is needed for the description in the methods.
3. The potential clinical values of the riskscore needs to be highlighted in the discussion.
4. Figure legends should be checked again. Some legends in figure 1 are missing.
5. The biological function of MYD88 in other types of cancer should be described and discussed in the discussion.
6. The description of Y axis in figure 4F and G are not consistent with the Y axis in figure 7A. I would suggest keeping it accordant.
7. The language needs to be improved. Some examples where the language could be improved include lines 49, 50, 65-73 – the current phrasing makes comprehension difficult. Please check the spelling mistakes in this manuscript.

Reviewer 3 ·

Basic reporting

A clear and professional English is used in this manuscript. Structure is consistent with PeerJ standards. Figures are high resolution and raw data is attached.

Experimental design

Research questions are well defined. The methods are suitable. Conclusions are accurate and rigorous.

Validity of the findings

The topic is meaningful in the field of glioma. The results and attached raw data can support the conclusions in the paper. It is meaningful to use single cell RNA sequencing analysis to enhance the novel functions of MYD88 in glioma cells.

Additional comments

The authors provided thorough analyses on a very important field of glioma. They identified differentially expressed ICD genes and developed a prognostic ICD-related model for glioma patients. The manuscript is nicely sectionalized, and the results of the analyses are well interpreted. The risk model built could be potentially useful in the prediction of glioma prognosis.
Major:
1. The authors did not do a good job in the introduction and did not address the topic well. The authors should think about how to introduce “glioma, immunogenic cell death, prognosis, therapies” in a better way.
2. One disadvantage is a vague description of the methods. Line 109-110, line 112-114
3. In figure 3, it is interesting that three clusters from ICD were matched with three types of immune phenotypes. Few discussion are presented about this finding.
4. In figure 6, it will enhance the results of figure 6C if the authors can provide the AUC values. line 214-222
Minor:
1. It will make sense to cite more articles for some parts. Line 41, 43, 62
2. A wrong label in figure 8B. Please check and revise it.

---

## Round 0.2 · accepted · Accept

The manuscript has been Accepted for publication.

Reviewer 1 ·

Basic reporting

no comment

Experimental design

no comment

Validity of the findings

no comment

Additional comments

no comment

Reviewer 2 ·

Basic reporting

no comments

Experimental design

no comments

Validity of the findings

no comments

Additional comments

no comments

Reviewer 3 ·

Basic reporting

No comments

Experimental design

No comments

Validity of the findings

No comments

Additional comments

No comments